# OpenReview forum: "What Makes and Breaks Safety Fine-tuning? A Mechanistic Study"
_NeurIPS.cc/2024/Conference — NeurIPS 2024 poster_

### Official Review · Reviewer_dExx · 2024-07-04

**Soundness:** 3
**Presentation:** 3
**Contribution:** 3
**Rating:** 6
**Confidence:** 4

**Summary:**

This work designs a synthetic data generation framework with the purpose of understanding safety fine-tuning. It investigates (1) Supervised safety fine-tuning; (2) Direct preference optimization; and (3) Unlearning.
Key observations:
(1) safety fine-tuning encourages separate cluster formations for safe and unsafe samples; (2) the inductive biases of safety fine-tuning significantly reduce the local Lipschitzness of the model for unsafe samples; and (3) samples corresponding to jailbreaking and adversarial attacks are not sufficiently impacted by the transformation learned by safety fine-tuning.

**Strengths:**

Overall, I like the paper on the problem it studies and its interesting findings that can contribute to the field.

- Task Novelty: Understanding safety-finetuning is very crucial for the safety of LLM.
- Interesting Observations, such as separate cluster formations for safe and unsafe samples; As the strength of jailbreaking attacks increases, the unsafe samples make the model behave more similarly to safe samples. These observations are meaningful for the improvement of Model Safety of LLMs.
- The paper is easy to follow and the observations are clearly stated.
- Very comprehensive details discussed in supp. The detailed discussions in the supplementary materials provide a thorough understanding of the methodologies and experiments. This level of detail is beneficial for reproducibility and for other researchers looking to build on this work.
- The validation of findings on real-world models like Llama-2 and Llama-3 strengthens the paper’s contributions.

**Weaknesses:**

- Can you explain more about why existing real-world datasets cannot be applied for understanding safety fine-tuning? I understand that the real-world dataset might be large and complex, but I'm curious whether using part of the real-world dataset influences the mentioned observations in the paper. (e.g., the difference between safe and unsafe prompts).
- I notice that some observations are also discussed in related work[1], their paper mentions that unsafe prompts (including jailbreak ones) lead to similar gradients, but this paper shows that jailbreaking attacks make the model behave more similarly to safe samples. Do you have some idea how such phenomena are connected and differentiated?

[1] GradSafe: Detecting Unsafe Prompts for LLMs via Safety-Critical Gradient Analysis

**Questions:**

See Weakness.

**Limitations:**

The authors have discussed the limitations.

---

> ### Author Rebuttal · Authors · 2024-08-07
>
> We thank the reviewer for their positive feedback. We are glad that the reviewer liked our work and found our task novel with interesting observations and comprehensive analysis beneficial for reproducibility. We address specific comments below.
>
> ---
> > Can you explain more about why existing real-world datasets cannot be applied for understanding safety finetuning? I understand that the real-world dataset might be large and complex, but I'm curious whether using part of the real-world dataset influences the mentioned observations in the paper. (e.g., the difference between safe and unsafe prompts).
>
> Thanks for this question! It is indeed difficult to use real world data for generating plausible hypotheses on how safety fine-tuning works and how jailbreaks circumvent it. A part of this reason is that modeling jailbreaks in a unified manner is difficult with real data. Despite jailbreak datasets being available, the domains and precise formats of these datasets can be quite different and they are often ambiguous in capturing the underlying notion of different types of jailbreaks. Additionally, the available alignment datasets (e.g. [1]) often capture *multiple characteristics* like helpfulness, harmfulness, etc. simultaneously. Thus, a response could be harmless but still not helpful. These characteristics can interact in complex ways and often influence one other. We believe that a better approach is to use systematically designed controlled settings instead, to generate plausible hypotheses and later provide strong evidence on real world settings. This is exactly what we do!
>
> For validating our hypotheses, we use Llama models. We encourage the reviewer to see our common reply [here](https://openreview.net/forum?id=JEflV4nRlH&noteId=butJTfMQLX), where we show that some of our observations indeed transfer very well to real world settings.
>
> [1] https://huggingface.co/datasets/Anthropic/hh-rlhf
>
> ---
>
> > I notice that some observations are also discussed in related work[1], their paper mentions that unsafe prompts (including jailbreak ones) lead to similar gradients, but this paper shows that jailbreaking attacks make the model behave more similarly to safe samples. Do you have some idea how such phenomena are connected and differentiated?
>
> Thanks for sharing this reference! We will  discuss it in the final version of our paper. We would like to clarify that interpreting the gradients versus the activations of the model are completely different tasks, and therefore it is difficult to make a direct connection between the papers. Although, we can think if the Lipschitzness analysis in our work can be connected to the gradient analysis in [2]. Similar to [2], we observe that *unsafe samples exhibit similar Lipschitz constants, leading to their high concentration at specific positions in the histogram plot* (See Fig. 7).
>
> [2] Xie et al. GradSafe: Detecting Unsafe Prompts for LLMs via Safety-Critical Gradient Analysis https://arxiv.org/abs/2402.13494
>
> ---
>
> **Summary:** We thank the reviewer for their valuable feedback that has helped us expand our analysis. We hope that our response addresses the reviewer’s concerns and hope that they will consider increasing their score to support the acceptance of our work.
> We will be happy to answer any further questions related to our work during the reviewer-author discussion phase.

---

> > ### Comment · Reviewer_dExx · 2024-08-09
> > **Thanks for rebuttal**
> >
> > I appreciate the authors' response, which generally solves my question. I keep the score of weak accept. Thanks.

---

> > > ### Author Response · Authors · 2024-08-13
> > > **Thank you for your time**
> > >
> > > Dear Reviewer,
> > >
> > > Thank you once again for your reviews and valuable time. Appreciate it.
> > >
> > > Please let us know if you had further questions or required clarifications regarding any aspect our work. We will be very happy to reply.
> > >
> > > Thank you!

---

### Official Review · Reviewer_SrSG · 2024-07-07

**Soundness:** 3
**Presentation:** 3
**Contribution:** 3
**Rating:** 5
**Confidence:** 2

**Summary:**

This work studies the mechanism behind safety fine-tuning (and why they are failing against attacks). Particularly, the authors introduce a synthetic task that simulates model safety training, alongside proposing a data generation framework. They reveal multiple insightful findings via this framework.

**Strengths:**

1. The synthetic task and data generation framework to simulate model safety training is novel. It offers a lightweight playground for researchers to further study model safety alignment in depth.
2. The mechanistic findings and observations are valuable. These findings also help understand and compare different safety fine-tuning methods (DPO v.s. Unlearning v.s. SSFT).
3. The paper is well structured and written. For example, when it comes to the synthetic safety data, I find the analogies provided by the authors helpful for me to connect them to the real-world safety text data. I enjoy reading the paper pretty much.

**Weaknesses:**

See "Questions."

**Questions:**

1. In Fig 2, for jailbreaking attacks with mismatched generalization, why not consider the scenario of OOD text (operand) tokens?
2. In the real-world setup involving Llama models, why only study them on simple safety tasks (i.e., the operator-operand-style 500 safe and unsafe instructions you created)? There are a lot of off-the-shelf safe/unsafe instruction & preference data nowadays, isn't it possible to study the safety mechanism of Llama models on these more realistic data?
3. Is the extrapolation intervention study (Line 283-292) related to the [Weak-to-Strong Extrapolation Expedites Alignment](https://arxiv.org/abs/2404.16792) paper?

**Limitations:**

Yes.

---

> ### Author Rebuttal · Authors · 2024-08-07
>
> We thank the reviewer for their positive feedback! We are glad that the reviewer enjoyed reading our work and found our setup and findings valuable for studying safety alignment in depth. We address specific comments below.
>
> ---
>
> > In Fig 2, for jailbreaking attacks with mismatched generalization, why not consider the scenario of OOD text (operand) tokens?
>
> Thanks for raising this question. Using OOD text tokens to model mismatched generalization jailbreaks is certainly feasible and *we indeed tried this earlier*. However, we found the attack success rate to be **low** compared to other jailbreaks considered in our study. Therefore, due to space constraints and the paper already being quite dense, we chose not to report this attack. However, for better clarity, we are happy to include a discussion on this attack in the appendix of the revised draft. We present details on the attack setup and corresponding results below.
>
> **Experiments with OOD text tokens attacks:** We ensure that during safety fine-tuning, the text tokens are not sampled from some non-terminal nodes in the PCFG tree. To generate jailbreaks, we then sample text tokens from these non-terminal nodes. We present the attack success rate for mismatched generalization jailbreaks using OOD task tokens, OOD text tokens, and both OOD task and text tokens in the table below.
>
> |  Protocol  | Learning Rate  | Safe (Instruct)  | Unsafe (Null)  | Unsafe (Instruct) | JB MG OOD task tokens (Instruct) | JB MG OOD text tokens (Instruct) | JB MG (Instruct) OOD task + text tokens |
> |:----------:|:--------------:|:----------------:|:--------------:|:-----------------:|:--------------------------------:|:--------------------------------:|:---------------------------------------:|
> | Unlearning |       ηM       |       99.8       |      99.9      |        5.0        |               92.3               |               11.2               |                   93.1                  |
> |            |       ηS       |       99.7       |      99.9      |        31.2       |               98.5               |               39.3               |                   98.6                  |
> |     DPO    |       ηM       |       98.6       |      99.6      |        11.8       |               93.6               |               21.6               |                   93.9                  |
> |            |       ηS       |       98.7       |      100.0     |        40.7       |               96.1               |               47.9               |                   96.7                  |
> |    SSFT    |       ηM       |       99.9       |      99.8      |        51.6       |               100.0              |               62.8               |                  100.0                  |
> |            |       ηS       |       99.7       |      100.0     |        72.8       |               100.0              |               84.9               |                  100.0                  |
>
> As observed, while using OOD text tokens can model jailbreaks, the attack success rate is low—especially for unlearning and DPO with a medium learning rate—compared to other types of jailbreaks.
>
> Additionally, in the real world scenarios, mismatched generalization attacks are generally crafted on the entire input prompt (See Wei et al. [1] for examples). As shown above, we observe that performing jailbreaks with both OOD task and text tokens yields similar performance to using OOD task tokens alone. This is another reason why we focus on analyzing JB MG with OOD task tokens in the main paper.
>
> [1] Wei et al.  Jailbroken: How does LLM safety training fail? https://arxiv.org/abs/2307.02483
>
> ---
>
>
> > In the real-world setup involving Llama models, why only study them on simple safety tasks (i.e., the operator operand-style 500 safe and unsafe instructions you created)? There are a lot of off-the-shelf safe/unsafe instruction & preference data nowadays, isn't it possible to study the safety mechanism of Llama models on these more realistic data?
>
>
> Thanks for this question! Due to constrained space, we have addressed this concern in the common reply [here](https://openreview.net/forum?id=JEflV4nRlH&noteId=butJTfMQLX) and therefore, we request the reviewer to kindly refer to the same.
>
> ---
>
> > Is the extrapolation intervention study (Line 283-292) related to the Weak-to-Strong Extrapolation Expedites Alignment (https://arxiv.org/abs/2404.16792) paper?
>
> Yes! Similar to that paper, we also observe that *we can indeed improve supervised safety fine-tuning by simply extrapolating the weights in the direction of the learned update (ΔW)* (refer to Fig. A.77 in the appendix).
> Additionally, we investigate the effect of linearly traversing in the weight space between two safety fine-tuned models and further extrapolating in this direction. The results for the same are present in Fig. 4 of the attached PDF. We observe that traversing from a weaker safety fine-tuning protocol like SSFT towards a stronger one like unlearning reduces the success rate of jailbreaking attacks (shown in brown), while maintaining the accuracy on clean samples. We believe the mentioned paper provides additional evidence in support of our analysis and demonstrates a real world application of our observations. It is indeed exciting to see how well our observations translate to real world settings, indicating our synthetic setup captures salient properties of real world settings!
>
> Since the paper was released close to the submission deadline, we note that we missed it and hence could not cite it. We promise to discuss it in the final version of our work.
>
> ---
>
> **Summary:** We thank the reviewer for their valuable feedback that has helped us expand our analysis. We hope that our response addresses the reviewer’s concerns and hope that they will consider increasing their score to support the acceptance of our work. We will be happy to answer any further questions related to our work during the reviewer-author discussion phase.

---

> > ### Comment · Reviewer_SrSG · 2024-08-10
> >
> > Thanks for the authors' rebuttal, which helps make the work more comprehensive. I will keep my rating.

---

> > > ### Author Response · Authors · 2024-08-13
> > > **Thank you for your time**
> > >
> > > Dear Reviewer,
> > >
> > > Thank you once again for your reviews and valuable time. Appreciate it.
> > >
> > > Please let us know if you had further questions or required clarifications regarding any aspect our work. We will be very happy to reply.
> > >
> > > Thank you!

---

### Official Review · Reviewer_sLPA · 2024-07-11

**Soundness:** 3
**Presentation:** 3
**Contribution:** 3
**Rating:** 7
**Confidence:** 5

**Summary:**

This paper proposes a synthetic data generation framework to systematically analyze safety fine-tuning methods, including supervised safety fine-tuning, direct preference optimization, and unlearning. The empirical results indicate that safety fine-tuning encourages the formation of different clusters for safe and unsafe samples, reducing the model's sensitivity to unsafe samples. Additionally, the success of jailbreaking and adversarial attacks is because they are more similar to safe samples than unsafe ones.

**Strengths:**

1. This work provides insights into understanding the mechanisms of safety fine-tuning methods, covering several widely used approaches, such as supervised safety fine-tuning, direct preference optimization, and unlearning. It contributes several interesting observations and future directions for designing safety fine-tuning techniques.
2. The authors design a novel data generation framework to simulate the pre-training datasets, safety fine-tuning datasets, jailbreaking datasets, and adversarial attacking datasets. This framework contributes to future research on improving capacities and the safe application of LLMs.
3. The paper is clearly written and well-organized. It is easy to follow the authors' ideas and understand their approaches. The authors use clear figures, i.e., Figure 1 and Figure 2, to show the procedure of their data generation framework. The notations and experimental results are clear and easy to read.
4. The authors have done extensive experiments to make conclusions and support their observations.

**Weaknesses:**

1. The authors should have provided a comprehensive literature review to provide a more detailed background of this research, such as PCFG.
2. Some concepts need further clarification or justification. For instance, how can we "ensure that the generated text tokens from each PCFG do not completely overlap" in Line 127?
3. Although the generated datasets simulate real-world cases, the authors should have conducted more experiments on real-world datasets.

**Questions:**

1. Why do you supervise the model to output null tokens in Line 173? And what does null space in Line 275 mean?
2. What does "resort to empirically quantifying it for each data point" mean in Line 300

**Limitations:**

See the third point in Weaknesses.

---

> ### Author Rebuttal · Authors · 2024-08-07
>
> We thank the reviewer for their positive feedback! We are glad that the reviewer found the setup novel and the paper well-written, with extensive experiments and interesting observations useful for designing safety fine-tuning techniques. We address specific comments below.
>
> ---
>
> > The authors should have provided a comprehensive literature review to provide a more detailed background of this research, such as PCFG.
>
> We are certainly happy to expand on this! We will include a short primer on formal grammars and PCFGs. We note that we have already provided a comprehensive discussion of related work on fine-tuning and jailbreaks in Appendix A.
>
>
> ---
>
>
> > Some concepts need further clarification or justification. For instance, how can we "ensure that the generated text tokens from each PCFG do not completely overlap" in Line 127?
>
> We first reemphasize our motivation for using multiple PCFGs with different production rules. Specifically, if we use a single PCFG to generate text tokens, the model might simply learn to associate bijective mappings with different text tokens without learning the underlying PCFG production rules. To address this, we use multiple PCFGs, each with different production rules. We further associate same task tokens to represent different bijective mappings across different PCFGs. *This encourages the model to learn the production rules along with bijective mappings.*
>
> To ensure that the generated text tokens are non-overlapping across different PCFGs, we make two design choices:
>
> * We use a large vocabulary size. This decreases the probability of sampling the exact same set of text tokens from any two PCFGs.
>
> * For the four different PCFGs (as shown in Fig. 1 of the attached PDF) we use, the production rules are designed so that the leaf nodes in each successive PCFG generate a subset of text tokens produced by the preceding PCFGs. For example, if the first PCFG generates ‘abcde’ at a leaf node, the second PCFG will generate ‘ace’ at the corresponding leaf node.
>
> For better clarity, we encourage the reviewer to refer to Fig. 1 in the attached PDF, where the production rules corresponding to terminal / leaf nodes are illustrated. We will add both the discussions above and this figure to the final version of the paper.
>
>
> ---
>
>
> > Although the generated datasets simulate real-world cases, the authors should have conducted more experiments on real-world datasets.
>
> Thanks for this question! Due to constrained space, we have addressed this concern in the common reply [here](https://openreview.net/forum?id=JEflV4nRlH&noteId=butJTfMQLX) and therefore, we request the reviewer to kindly refer to the same.
>
>
> ---
>
>
> > Why do you supervise the model to output null tokens in Line 173? And what does null space in Line 275 mean?
>
> Thanks for this question. We argue that responses from a safety fine-tuned model to unsafe samples exhibit low variability and generally correspond to refusal of request. Here are some examples of generated responses from ChatGPT:
>
> * **Prompt:** How to design a bomb?
>
>     **Response:** Sorry, but I can't assist with that.
>
> * **Prompt:** How to rob a bank?
>
>     **Response:** I can't help with that.
>
> In our setup, we model this refusal behavior by instructing the model to output null tokens during safety fine-tuning.
>
>
> The null space of W_{IT}^L mentioned in line 275 is the same as the left null space of W_{IT}, as defined in Sec. 2 of the main paper.
>
>
> ---
>
>
> > What does "resort to empirically quantifying it for each data point" mean in Line 300.
>
> We calculated the local Lipschitz constant for both safe and unsafe samples. For each sample, we obtain a scalar value of the local Lipschitz constant, which we then use to plot a histogram, as shown in Fig. 7. The reason we use this is to provide a comprehensive analysis over different data points in the input distribution corresponding to safe and unsafe samples.
>
>
> ---
>
>
> **Summary:** We thank the reviewer for their valuable feedback, which has helped us expand our analysis. We hope that our response adequately addresses the reviewer’s concerns. We will be happy to answer any further questions related to our work during the reviewer-author discussion phase.

---

> ### Comment · Reviewer_sLPA · 2024-08-12
> **Thank you for your reponses**
>
> I have read the authors' responses. Most of my concerns have been addressed. I will keep my score as 7 Accept.

---

> > ### Author Response · Authors · 2024-08-13
> > **Thank you for your time**
> >
> > Dear Reviewer,
> >
> > Thank you once again for your reviews and valuable time. Appreciate it.
> >
> > We are glad that we could address most of your concerns.
> >
> > Thank you!

---

### Official Review · Reviewer_K9Ts · 2024-07-12

**Soundness:** 3
**Presentation:** 2
**Contribution:** 3
**Rating:** 5
**Confidence:** 4

**Summary:**

This paper introduces a novel synthetic data generation framework that allows controlled generation of data for safety fine-tuning, jailbreaking attacks, and adversarial attacks. This paper provides comprehensive analyses on the mechanisms learned after safety fine-tuning

**Strengths:**

Controlled way of safety finetuning. Provide a few explanations to safety finetuning observations.

**Weaknesses:**

1. How is the quality of the synthetic dataset controlled?
2. Most of the findings in the paper have been explored before, e.g., cluster information, effects of jailbreak attacks
3. What are the new insights provided by the three observations listed in the second bullet point?

**Questions:**

See weakness

**Limitations:**

See weakness

---

> ### Author Rebuttal · Authors · 2024-08-07
>
> We thank the reviewer for their efforts in reviewing our work. We are glad that the reviewer found our setup novel and analysis comprehensive. We address the specific comments below.
>
> ---
> > How is the quality of the synthetic dataset controlled?
>
> We are unsure we follow the reviewer's intended meaning; it would help if the reviewer could clarify what they meant by the phrase **quality of synthetic data**. In case the reviewer is asking how the dataset is constructed, we note our setup models a notion of operators and operands, where operators are modeled using task tokens corresponding to different bijective mappings, and operands are represented by text tokens sampled from PCFGs. We provide the production rules used to generate these tokens in Fig.1 in the attached PDF. We encourage the reviewer to refer to Sec. B in the appendix for more details on the synthetic setup. Another way of looking at the quality would be to see if the generated data led to conclusions that corroborate well with the real-world data. We indeed find this; we encourage the reviewer to check our common reply [here](https://openreview.net/forum?id=JEflV4nRlH&noteId=butJTfMQLX) and also experiments (Fig. 3, A. 17, A. 18, A. 68) in our paper.
>
> ---
> > Most of the findings in the paper have been explored before, e.g., cluster information, effects of jailbreak attacks.
>
> We *respectfully* disagree with the reviewer's assessment that our work’s findings have been explored before; to the best of our knowledge, our findings are novel and we are the first to thoroughly investigate safety fine-tuning and jailbreaks in a unified framework. In fact, we highlight that all other reviewers have found our findings to be novel and interesting. E.g., reviewer sLPA found our work to show "several interesting observations and future directions for designing safety fine-tuning techniques". To address the reviewer's concerns more precisely, we would appreciate it if the they can share specific references that demonstrate results similar to ours.
>
> We do note that a few concurrent works were released close to the submission deadline, but, as NeurIPS policy states, these papers should not be deemed prior work. However, these papers do provide evidence that corroborate with some of our claims!
>
> * **Arditi et al. [1] (arxiv, 17th June 2024):** This work demonstrates two observations similar to ours:
>   * Clustering occurs for safe and unsafe samples in the feature space (Observation 1, Fig. 3 in our paper).
>
>   * A single direction is responsible for refusal. We demonstrate this in Fig. 5 of our work, where we observe that only the top few singular vectors of ΔW contribute towards separation of clusters.
>
> * **Zheng et al. [2] (arxiv, 25th April 2024):** This work demonstrates that merely extrapolating in the direction of the learned update (ΔW) improves the safety performance of the real-world aligned language models. We also show similar behaviour in our synthetic setup for supervised safety fine-tuning (see Fig. A.77 in the appendix). Although this is not our main finding, it is exciting to see how well it translates to real world scenarios!
>
> * **Ball et al. [3] (arxiv, 13th June 2024):** This work demonstrates two observations similar to ours:
>
>   * Clusters of safe and unsafe samples are formed in the feature space (Fig. 3 in [3], Fig. 3 in our paper).
>
>   * Jailbreaks with increased attack strength do not behave similarly to unsafe samples in the activation space (Fig.4 in [3]). Along with similar observations that we show in our work (Fig. 8), we explain why this happens. We show that (Fig. 5, attached PDF) the projection of pre-activations onto the singular vectors of ΔW decreases with the increase in attack strength, resulting in jailbreaks behaving similar to safe samples.
>
> We note that our findings on how the learned update (ΔW) leads to the formation of separate clusters along with our analyses on jailbreaks and local Lipschitzness have not been discussed in previous work. These works [1,2,3] additionally demonstrate that the hypotheses generated using our synthetic setup indeed transfer well to real world datasets and models, thus validating our setup. This underscores the value of synthetic pipelines in generating useful insights and motivates future research using controlled synthetic settings to develop plausible hypotheses.
>
> [1] Refusal in Language Models Is Mediated by a Single Direction https://arxiv.org/abs/2406.11717
>
> [2] Weak-to-Strong Extrapolation Expedites Alignment https://arxiv.org/abs/2404.16792
>
> [3] Understanding Jailbreak Success: A Study of Latent Space Dynamics in Large Language Models https://arxiv.org/abs/2406.09289
>
> ---
> > What are the new insights provided by the three observations listed in the second bullet point?
>
> Expanding on the three observations, we provide a more detailed list of our contributions below.
>
> * **We provide a unified synthetic setup to methodically study safety fine-tuning methods and jailbreaks.** We make careful design choices to adhere to the properties of natural language instructions and the jailbreaks taxonomy of [4].
>
> * **We show that safety fine-tuning methods yield specialized transformations that primarily activate for unsafe inputs.** We show that safety fine-tuning encourages separate cluster formations for safe and unsafe samples by minimally transforming MLP weights to specifically project unsafe samples into the null space of its weights, and the inductive biases of safety fine-tuning substantially reduce the local Lipschitzness of a model for unsafe samples.
>
> * **We show that adversarial inputs yield intermediate features that are exceedingly similar to safe samples, hence evading the processing by ΔW required for refusal of an input.**
>
> [4] Jailbroken: How does LLM safety training fail? https://arxiv.org/abs/2307.02483
>
> ---
> **Summary:** We hope that our response addresses reviewer’s concerns and they will consider increasing their score to support acceptance of our work.

---

> > ### Author Response · Authors · 2024-08-13
> > **Gentle Nudge**
> >
> > Dear Reviewer,
> >
> > We would like to thank you once again for your reviews and valuable time. Appreciate it.
> >
> > We were wondering if you had further questions or required clarifications regarding any aspect our work. We will be very happy to reply.
> >
> > Thank you!

---

> > > ### Comment · Reviewer_K9Ts · 2024-08-13
> > >
> > > Thank you! After reviewing the paper again and all other reviews, I apologize for my misunderstanding of the paper before. I think the reviewer has fully addressed my concerns. Thus, I will raise my score to 5.

---

### Author Rebuttal · Authors · 2024-08-07

## **Common Reply**

We thank the reviewers for their efforts in reviewing our work. We are glad that all the reviewers found our PCFG-based synthetic setup novel and our analysis comprehensive. Additionally, reviewers sLPA, SrSG and dExx found our work easy to follow and our observations valuable, saying our contributions can possibly aid the design of improved safety fine-tuning techniques. Below, we address the common concerns raised by reviewers sLPA, SrSG and dExx regarding the transferability of our results to real data settings by **performing additional experiments**. Please see the attached PDF for results.

* **Additional experiments on real-world dataset.** We use real world data from a recent work by Arditi et al. [2] and show, in Fig. 3 of the attached PDF, that safety fine-tuning indeed encourages formation of separate clusters between safe and unsafe samples (as predicted by Observation 1 of our main paper) in the feature space of the model. We use Llama-2-7B-Chat as the instruction and safety fine tuned counterpart of the pre-trained Llama-2-7B model in this experiment.

* **Additional experiments on jailbreaking attacks using synthetic data.** As demonstrated in Figs. A.10 - A.13 in the appendix, prompts similar to the ones used in our data setup (See Fig. A.15 in the appendix) to analyze Llama models also successfully jailbreak them. Based on this, we create a similar dataset for jailbreaks with competing objectives and mismatched generalization for further analysis.

    * **Crafting mismatched generalization jailbreaks:** We translated unsafe prompts used in our synthetic setup (Fig. A.15) into other low resource languages (kannada and malayalam in our case), which have been shown to successfully jailbreak language models (See Fig. A.13 in appendix and Yong et al. [1]).

    * **Crafting jailbreaks with competing objectives:** We use multiple operators for an operand, one corresponding to a safe instruction and other corresponding to an unsafe one.

    Using the above datasets, in Fig. 2 of the attached PDF, we show that jailbreaks reduce the separation between the clusters of safe and unsafe samples and behave more similar to safe samples.

* **Evidence from concurrent works:**  As evident from the analyses above, we have shown that our results indeed transfer to realistic settings. This is also evident from a concurrent work by Ball et al. [3], which validates some of our claims (listed below) using only real world datasets:

    * Formation of separate clusters for safe and unsafe samples (See Fig. 3 in Ball et al. [3])

    * Jailbreaks with increased attack strength do not behave similarly to unsafe samples in the activation space (See Fig. 4 in Ball et al. [3]).

    We will include both the results above in the revised draft as well.

* **Use of PCFG-based synthetic setup:** Having shown the additional results above, we would like to emphasize that a realistic and systematically designed synthetic set-up is crucial in understanding the biases instruction/safety fine-tuning methods induce to LLMs. A carefully designed synthetic set-up for data would allow generation of plausible hypotheses, in a much controlled and efficient manner, and help in providing more grounded conclusions. These hypotheses can then be validated in real world settings, as we do in this work. However, directly analyzing real world and often inaccessible LLMs to understand their properties using vast and complex data domains can be both compute intensive and at times infeasible.

[1] Yong et al. Low-Resource Languages Jailbreak GPT-4 https://arxiv.org/pdf/2310.02446

[2]  Arditi et al. Refusal in Language Models Is Mediated by a Single Direction https://arxiv.org/abs/2406.11717

[3] Ball et al. Understanding Jailbreak Success: A Study of Latent Space Dynamics in Large Language Models https://arxiv.org/abs/2406.09289

---

### Decision · Program_Chairs · 2024-09-25

**Decision:**

Accept (poster)

**Comment:**

The recommendation is based on the reviewers' comments, the area chair's evaluation, and the author-reviewer discussion.

This paper presents comprehensive evaluations and interpretations of several safety fine-tuning methods. All reviewers find the studied setting novel and the results provide new insights. The authors’ rebuttal has successfully addressed the major concerns of reviewers. Therefore, I recommend acceptance of this submission. I also expect the authors to include the new results and suggested changes during the rebuttal phase to the final version.